# Changes in corneal thickness after vitrectomy—Implications for glaucoma practice

**Lisika Gawas, Aparna Rao** *

Glaucoma Services, LV Prasad Eye Institute, Bhubaneswar, India

* vinodini10375@yahoo.com

## Abstract

### Purpose

To evaluate changes in central corneal thickness (CCT) following vitrectomy.

### Methods

All consecutive old and new patients referred to glaucoma services for possible secondary glaucoma after vitrectomy and who had undergone corneal pachymetry between July 2013 to June 2020, were included. The eye that developed elevated intraocular pressure (IOP) and was diagnosed clinically as glaucoma after vitrectomy, was labelled as the "affected" eye. The contralateral eye of the patient with normal IOP and no history of vitrectomy was labelled as the "control" eye. The difference in CCT in the affected eye and the contralateral control eye (ΔCCT) and CCT were compared between different age groups. Correlation of CCT in the affected eye with age, diagnosis, type of surgery done, lens status and pre-existing glaucoma was done using multivariate regression analysis.

### Results

Of 127 eyes of 120 patients (M:F = 85:35), the average CCT in the affected eye was significantly higher than the unaffected contralateral control eye ($p < 0.0001$). The ΔCCT in eyes presenting at an age <25 years was higher (median 582, 497–840) than those that presented later (median 518, 384–755), $p < 0.0001$, with maximum ΔCCT seen in eyes that had undergone vitrectomy at age<12 years. The CCT in the affected eye was significantly higher in aphakic eyes (588±81.6 microns) than in pseudophakic eyes (552±79.03 microns), $p = 0.03$. On multivariate analysis, age<25 years remained as a significant influencer of CCT in the affected eye (β = -1.7, $p < 0.001$, $R^2$ = 28.3%).

### Conclusions

Young age group<25 years are more prone to corneal remodelling and CCT changes after vitrectomy.

**Data Availability Statement:** All relevant data are within the manuscript and its Supporting Information files.

**Funding:** The authors received no specific funding for this work.

**Competing interests:** The authors have declared that no competing interests exist.

## Introduction

Central corneal thickness (CCT) has been recognized as a significant risk factor for the progression of ocular hypertension to primary open-angle glaucoma in the Ocular Hypertension Treatment Study [1, 2]. It is known from several studies that intraocular pressure (IOP) may be influenced by CCT or corneal biomechanical properties, which may indirectly parallel the biomechanical properties of the optic nerve head [3, 4]. Though the role of CCT in glaucoma pathogenesis is debatable, it remains an invaluable tool in routine glaucoma practice [3, 5, 6]. This is especially important in situations where we want to accurately interpret Goldman applanation tonometry (GAT) readings to avoid over or under-treatment of glaucoma [3, 5]. Corneal remodelling or changes in CCT have been reported after anterior and posterior segment surgeries [7–11]. While these changes are commonly encountered after refractive surgeries, its application in glaucoma assumes importance after posterior segment surgeries where post-operative corneal remodelling may cause false diagnosis of raised intraocular pressure (IOP) or secondary glaucoma [3, 4, 7, 9].

Pars plana vitrectomy (PPV) is one of the most frequently performed ophthalmic surgeries used in the treatment of many vitreous and retinal diseases. Elevated IOP and progressive glaucomatous damage are known postoperative complications following vitreoretinal surgeries [8–10]. Secondary IOP elevation after pars plana vitrectomy with silicone oil injection has been reported in 5.9% to 56% of eyes [9, 11, 12–14]. While it is important to identify glaucoma after vitreoretinal surgeries and initiate early treatment, it is also known that CCT may change after anterior and posterior segment surgeries. This study evaluates long-term changes in CCT after vitrectomy with its direct implication in the diagnosis of glaucoma.

## Method

All consecutive old and new patients referred to the glaucoma services for possible secondary glaucoma after vitrectomy (which included anterior automated vitrectomy, AV+ 23-G pars plana vitrectomy, PPV) between July 2013 and June 2020, were identified from the hospital electronic medical records database. The study was part of a retrospective study comparing outcomes of primary and secondary glaucoma, that was approved by the Institutional Review Board of LVPEI, MTC campus, Bhubaneswar, and adhered to the tenets of the declaration of Helsinki. Of these, only patients who were advised corneal pachymetry (CCT) measurement, were included for this study. Demographic data extracted from the electronic medical records included age, gender, presence of pre-existing glaucoma, best-corrected visual acuity (BCVA), IOP, lens status, +90 D fundus biomicroscopy, final diagnosis, type of prior surgery before presentation to glaucoma services (anterior or posterior segment procedures included), medical or surgical management, and final outcomes. The need for additional interventions after initial glaucoma management was also obtained from the records.

By institutional protocol, all IOP readings are obtained by trained optometrists using Goldman applanation tonometer(GAT) under local anesthesia. After the patient is sitting comfortably at the slit lamp: at the right height, with their chin on the rest and their forehead against the headband, the cornea is stained with fluorescein, and measurements are taken under the cobalt blue filter with the patient fixing straight ahead. The calibrated dial on the tonometer is turned clockwise until the inner edges of the two fluorescein semi-circles touch each other. Pachymetry was done by a single technician under topical anaesthesia using ultrasound pachymetry(model- SP 3000, Japan). The probe is placed on the central portion of the cornea with the patient seated comfortably and looking straight. An average CCT is obtained after a total of 6 readings with a standard deviation between readings <5 microns. Any variability in readings with large standard deviations in irregular corneas, and uncooperative patients were excluded. All pars plana

vitrectomy surgeries were done by trained surgeons under local anaesthesia, with the infusion cannula at the inferotemporal quadrant and non-contact wide-field system being used for visualization. A posterior vitreous detachment was induced in cases with an intact hyaloid. After surgery, the ports were closed with 8–0 vicryl sutures, ensuring absence of vitreous incarceration in the wound after postoperative screening of the retina before closure.

For patients with possible secondary glaucoma post-vitrectomy, glaucoma was diagnosed in the presence of >21mm Hg IOP at any visit, with no prior history of raised IOP or anti-glaucoma medications (AGM). Glaucomatous disc and corresponding visual field changes at presentation were not mandatory for a diagnosis of glaucoma in cases with no prior history of glaucoma. Medical management was initiated and surgery was advocated for uncontrolled IOP despite maximum tolerated medical therapy. Any eye that developed a raised IOP and was clinically diagnosed as glaucoma after vitrectomy was labelled as the "affected" eye, while the contralateral eye of the patient with normal IOP and no prior history of vitrectomy, was labelled as the "control" eye. If both the eyes of the same patient had undergone vitrectomy, both eyes were labelled as affected. The CCT in the affected eyes and control eyes were compared in eyes that underwent anterior and posterior vitrectomy. The difference in CCT in the affected eye and the contralateral control eye (hitherto referred to as ΔCCT) indirectly represents the amount of corneal remodelling resultant to stress or surgery (A direct ΔCCT is ideally obtained for any eye by pre and post-operative CCT measurements, which is not done routinely in for all eyes undergoing vitrectomy; this makes pre-operative CCT measurements comparisons difficult).The ΔCCT was compared in different age groups and in eyes undergoing different routes of vitrectomy (anterior versus posterior).The final diagnosis after completion of glaucoma investigations (CCT, IOP, fundus, and visual fields, if possible) and management outcomes in the affected eye and unaffected control eye were also analysed.

## Statistics

All analysis was done using StataCorp (version 11, USA). Descriptive data are presented as mean+standard deviation (or median and interquartile range) with statistical significance set at $p < 0.05$.Normality was checked using the Shapiro-Wilk test. Comparison of CCT between the control and affected eye was done using paired t-test. The CCT and ΔCCT in the affected eye across different age groups and eyes undergoing different surgeries(pars plana vitrectomy or anterior vitrectomy)were compared. The CCT in eyes with and without a final diagnosis of glaucoma(those that were observed and presumed normal by the clinician) was compared using the independent student -t test. Relationship of CCT in the affected eye with age, diagnosis, type of surgery done, lens status, pre-existing glaucoma, and the relationship of ΔCCT with the final diagnosis of glaucoma or need for surgery/additional management was done using multivariate regression analysis.

## Results

Of 286 patients presenting to the glaucoma service post-vitrectomy (anterior+pars plana),127 eyes of 120 subjects (M:F = 85:35) who had undergone pachymetry were included for this study, with a median age of 41 years (range 7–85 years) and a mean presenting IOP of 27± 11.01mmHg. Of these, 4 patients had undergone surgery in both eyes with one eye of a patient with no vision being excluded (127 affected eyes, 120 control eyes). The indications for surgery were retinal diseases (70.09%), dislocated/subluxated lens (16.54%), vitreous in the anterior chamber (11.81%) and macular disease (2.37%) with maximum affected eyes being aphakic (n = 50) or pseudophakic (n = 60). The most common surgery done in the patients was retinal detachment (42.52%) followed by diabetic vitreous haemorrhage (12.6%). Of 127 eyes, 113

eyes had a final diagnosis of glaucoma after all appropriate investigations while 14 were deemed as normal by the clinician at presentation. At the final visit, a total of 115 eyes (this included 6 eyes that were diagnosed as normal at presentation and later initiated on medicines on follow-ups for IOP control)required antiglaucoma medications, while12 eyes underwent glaucoma surgery for IOP control.

## Affected eye comparisons

The average CCT in the affected eye was significantly higher than the unaffected contralateral control eye (p<0.0001), **Table 1**, **Fig 1**. The CCT in the affected eye was significantly higher in aphakic eyes (588±81.6 microns, IQR = 472–840 microns) than in pseudophakic (552±79.03 microns, IQR = 384-902microns) or phakic (532±48.4, IQR = 471–515 microns) eyes (p = 0.03). This correlated with maximum ΔCCT in aphakic eyes, **Fig 2**.

Comparing the age-wise distribution of CCT in the affected eye, eyes undergoing vitrectomy at a younger age <25 years had thicker corneas (603±77.6 microns) as compared to those that underwent surgery at a later age (543±60.5), **Table 2**, p<0,001. This also paralleled with a greater ΔCCT in eyes presenting at an age<25 years, (median 582, 497–840) compared to those that presented later (median 518, 384–755),p<0.0001.Though the mean CCT was not significantly different in eyes undergoing anterior or pars-plana vitrectomy, the CCT was thickest in eyes <25 years, with maximum ΔCCT seen in eyes undergoing vitrectomy at age<12 years, **Table 2**. Those >40 years showed the least difference between the affected and normal eye, **Table 2**. Comparing eyes with or without pre-existing glaucoma, the CCT and ΔCCT were not statistically different between the two groups, **S1 Table**. There was no correlation of ΔCCT or extent of corneal remodelling to the need for additional surgery at follow-ups.

Parallel to the CCT, the presenting IOP for eyes undergoing PPV was higher than those presenting after anterior vitrectomy, **Table 3**. This was seen in eyes <25 or >25 years signifying a minimal effect of age on the presenting IOP. Though thepresenting IOP was higher in eyes after PPV, the number of eyes requiring AGM, those that had disc/field progression did not differ across age groups or type of surgery, **Table 3**.

Of 14 eyes that were observed, 6 eyes showed raised IOP >30mmHg over a mean follow-up period of 27±22.5 months and 4 of 6 eyes showed disc/field progressionmandating treatment. All the 6 eyes initiated on medical treatment (mean number of medicines = 1.5±0.4, with none of the eyes requiring more than 2 medicines) had adequate IOP control with none of the eyes requiring surgery.

Univariate analysis of CCT and ΔCCT showed significant association with age<25 years (r = -0/4, p<0.001) and aphakic lens status(r = 0.2, p = 0.02). In multivariate analysis, only age<25 years remained as a significant influencerof CCT in the affected eye (β = -1.7,

**Table 1. Differences in baseline characteristics of affected eyes and normal eyes undergoing vitrectomy.**

| Variables | Affected Eye* | Normal Eye | P value# |
|---|---|---|---|
| | N = 127 | N = 120 | |
| Presenting IOP (mmHg) | 27 ± 11.01 | 14 ± 4.37 | <0.0001 |
| Visual acuity (logMAR) | 1.6 ± 0.75 | 1 ± 2.91 | 0.03 |
| CCT (microns) | 566 ± 78.29 | 528 ± 41.17 | <0.0001 |
| Duration of follow up (months) | 27.32±10.5 | 27.32±17.6 | 0.6 |

IOP-intraocular pressure; CCT-central corneal thickness

*-see text for definition of affected eye and normal eye

#-paired t test done for 120 eyes only-see text for full description.

## A-affected eye

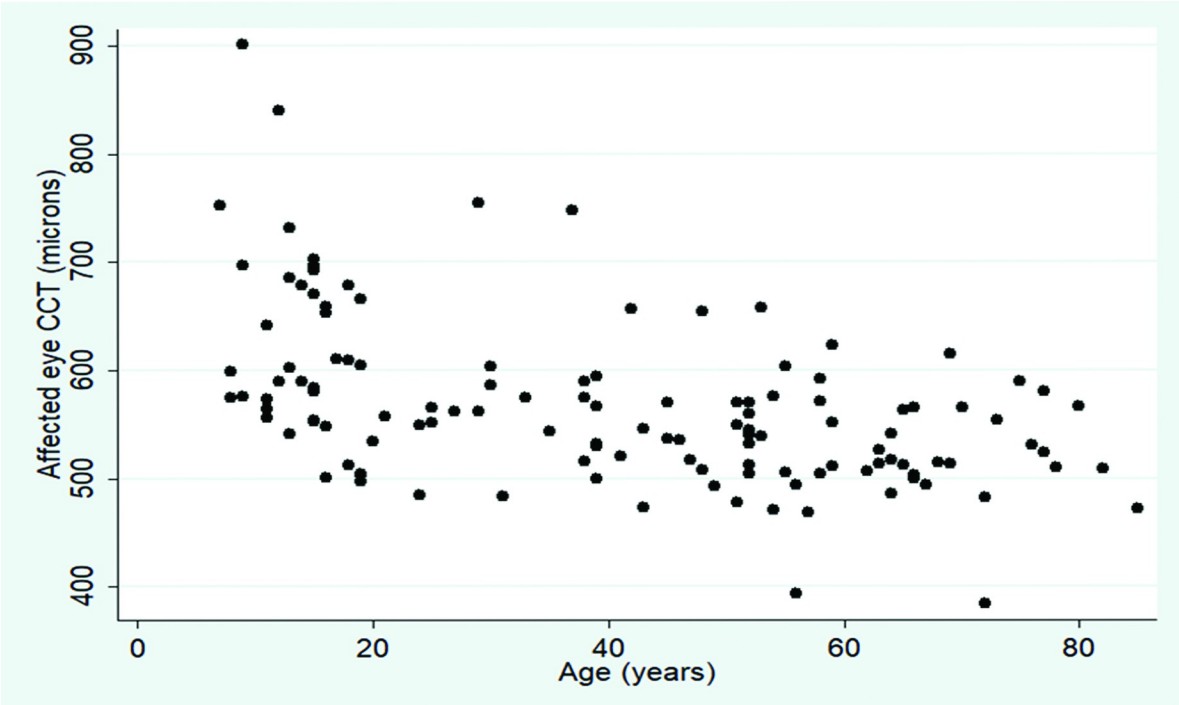

## B-Normal eye

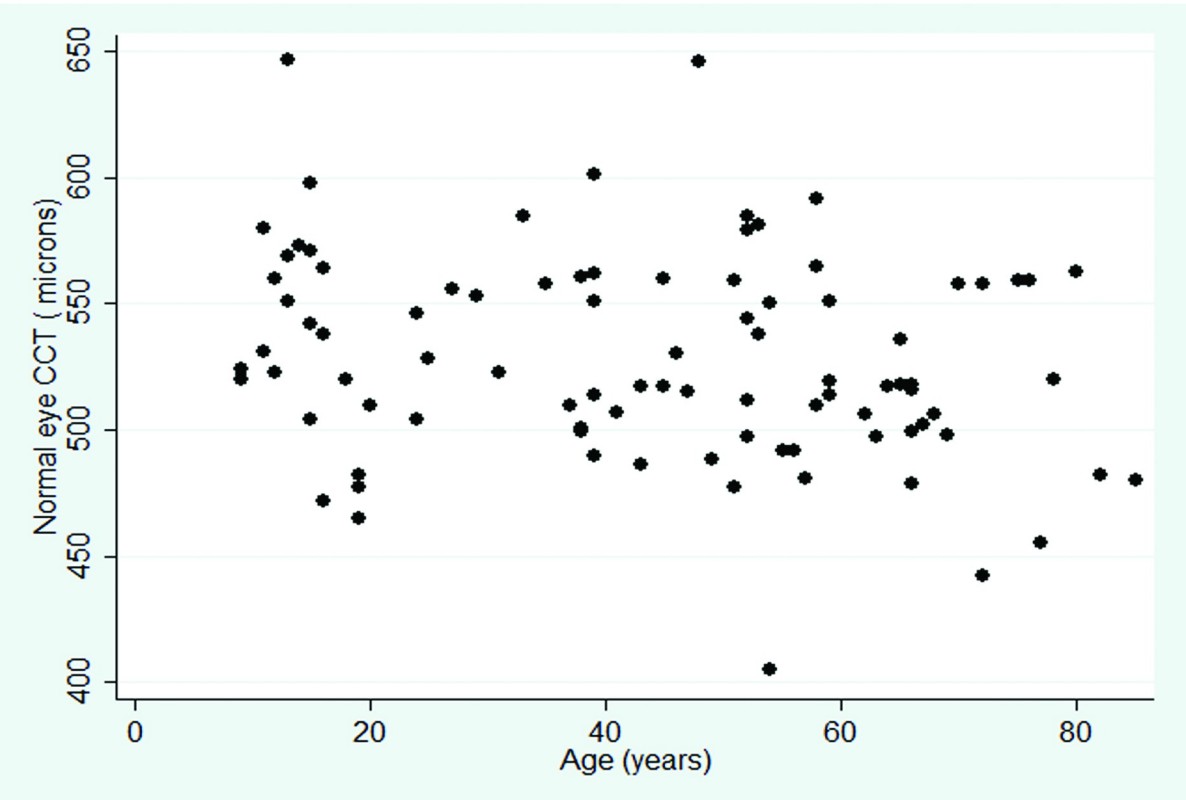

**Fig 1. A-shows the central corneal thickness in affected eye undergoing vitrectomy while B shows the corneal thickness in the contralateral normal eye (see text for detailed definition of affected and normal eye).**

p<0.001, R2 = 28.3%) and ΔCCT(β = -0.7, p = 0.01, R2 = 17.4%) with no influence of the lens status, type of surgery (PPV versus AV) or presenting IOP, gender, pre-operative diagnosis or presence of pre-existing glaucoma.

## Discussion

This study found thicker CCT in affected eyes after vitrectomy, with greater ΔCCT seen in aphakic eyes and in age<25 years. Among <25 years, this change was maximally seen in the paediatric age group<12 years. There was no statistically significant difference in CCT in eyes undergoing pars plana vitrectomy and anterior vitrectomy. The CCT and ΔCCT was significantly influenced by the age at which vitrectomy was done, with the type of surgery or baseline IOP or even pre-existing glaucoma not influencing the ΔCCT, an indirect measure of corneal remodelling after vitrectomy.

The significance of corneal thickness in glaucoma though established in routine clinical practice was first highlighted by the OHTS trial [1–3]. Several studies have reported the utility of CCT in glaucoma [3, 15–20]. Measurement of CCT remains an important tool in routine clinical glaucoma practice in identifying over or under-estimation of GAT readings and titrate target IOP range with medical therapy [3, 5, 6, 16]. The Early Manifest Glaucoma Trial (EMGT) found that thinner CCT (1.01–1.55 per 40 mm lower) may play a role in predicting progression in those with higher baseline IOP [15]. Gordon et al found that participants in the OHTS with a CCT <555 microns had a 3 fold greater risk of developing glaucoma [1]. The European glaucoma prevention study reported that CCT could be an independent risk factor for the development of open-angle glaucoma [17]. Though we understand CCT as a significant risk factor for glaucoma, it is postulated that corneal properties and stiffness mirror that of the lamina cribrosa in an indirect fashion rather than having a direct influence on IOP or glaucoma [18–20].

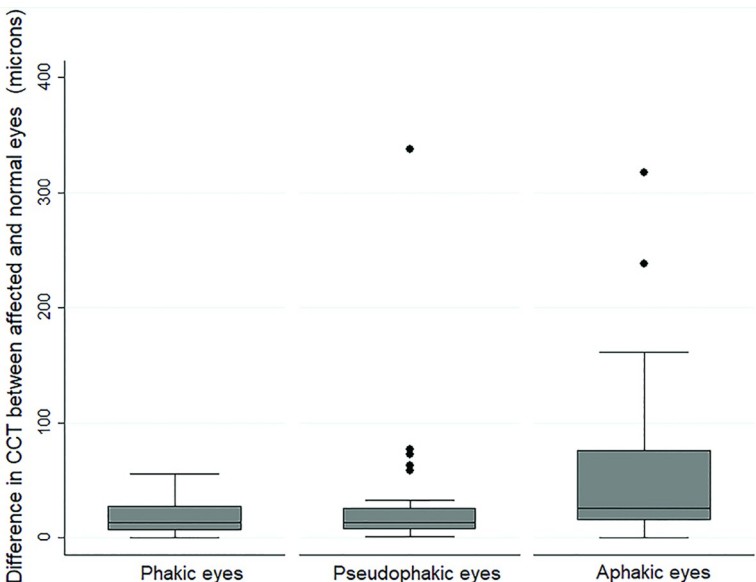

**Fig 2. Boxplot of Δ central corneal thickness (CCT) after vitrectomy in eyes with different lens status showing maximum ΔCCT in aphakic eyes. (See text for description of ΔCCT).**

**Table 2. Comparison of central corneal thickness and ΔCCT in eyes undergoing anterior and posterior vitrectomy (See text for details of ΔCCT description and significance levels of comparisons).**

| Type of surgery | Age | Affected eye (Microns) | Normal eye (Microns) | ΔCCT (Microns) |
|---|---|---|---|---|
| | | Mean±SD | Mean±SD | Mean±SD |
| Pars Plana Vitrectomy | ≤12 | 647± 130.2 | 535 ± 25.1 | 338±109.9 |
| | 13–25 | 587±70.35 | 536.± 48.9 | 161±52.2 |
| | 26–40 | 574± 68.8 | 544 ± 34.9 | 123±66.5 |
| | >40 | 532±53.3 | 521±42.6 | 125±21.9 |
| Anterior vitrectomy | ≤12 | 680 ± 82.81 | 560+10.1 | 218±21.1 |
| | 13–25 | 607± 74.2 | 523±57.9 | 63±10.6 |
| | 26–40 | 607±96.8 | 528±27.1 | 238±49.6 |
| | >40 | 529±36.7 | 517 ±33.5 | 31±8.4 |

SD-standard deviation; CCT-central corneal thickness-see text fr explanation for ΔCCT.

A raised IOP after retinal interventions can be due to secondary glaucoma by various causes or simply due to these changes in CCT owing to corneal remodeling [3, 10, 11, 14]. Ascertaining the diagnosis of a disease is therefore particularly important before initiating glaucoma treatment after such surgeries where both factors may be seen together. Conventionally, a cause for secondary glaucoma after retinal surgeries entails investigating type of surgery and possibly pre-existing disease. A change in CCT causing the raised IOP is often missed in routine glaucoma practice. An increased CCT after any surgery may therefore reflect a change in the stromal thickness by stress-induced keratocyte activation or changes in the endothelial density [8, 11, 13, 14, 20–29]. It is also postulated that changes in corneal curvature or axial length by retinal interventions may also contribute to changes in the corneal biomechanics or CCT [8, 9, 21, 22]. While stromal changes are commonly reported after refractive surgery, similar studies done after retinal interventions are fewer [3, 8]. Hager et al showed a significant increase in CCT after 20-G pars plana vitrectomy (34.95 ± 23.57 microns) and cataract surgery (23.76 ± 26.0 microns) [7]. In contrast, Seymenoğlu et alreported no change in CCT after 23-G vitrectomy, signifying the effect of improved surgical technique and lesser surgical stress with 23-G compared to 20-G vitrectomy [21]. Mukhtar et al reported minimal changes in CCT 6months after 23-G PPV or buckle surgery [25]. Watanabe et al reported an immediate increase in CCT after vitrectomy with a direct correlation of the change in CCT to the severity of surgical stress [8]. Other studies have reported long-term (3 months) endothelial density changes after RD surgeries or after PPV. These changes have been reported to be maximum in

**Table 3. Comparison of age-wise distribution of central corneal thickness in the affected eye and control eye after pars plana vitrectomy or anterior vitrectomy.**

| | Age <25 years undergoing PPV | Age >25 years undergoing PPV | Age<25 years undergoing AV | Age>25 years undergoing AV | P value |
|---|---|---|---|---|---|
| | N = 33 | N = 72 | N = 10 | N = 12 | |
| Mean presenting IOP (mm Hg) | 29 ± 9.9 | 28 ± 10.5 | 25±15.6 | 20±8.7 | 0.02 |
| Number of eyes on AGM at final visit | 32 | 72 | 10 | 12 | 0.6 |
| Requirement for glaucoma surgery | 2 | 8 | 1 | 1 | 0.9 |
| Disc progression | 0 | 5 (3 with VF progression) | 0 | 1 | 0.3 |
| Field progression | 0 | 5 | 0 | 1 | 0.3 |

IOP-Intraocular pressure; AGM-anti-glaucoma medications.

pseudophakic and aphakic eyes, suggesting the importance of the lens in protecting the endothelium [8–10]. Similar changes have been reported after the use of silicon oil or scleral buckle surgery [12–14, 22, 23–25]. These studies contrast with our study where long-term CCT changes were seen maximally in the younger age groups. This was presumably owing to a different study design, the inclusion of cases that were seen in the glaucoma services for raised IOP after surgery, and the direct application of CCT in glaucoma and its treatment. It may be argued that no pre-operative CCT was done in the affected eye and the contralateral eye was taken as the control eye. Yet, we do not believe that inter-eye difference in CCT would be significantly different in any patient until surgery is undertaken in any eye. Knowledge about the change in CCT after vitrectomy causing a raised IOP can avoid over-diagnosis of glaucoma in these eyes referred to the glaucoma clinic for treatment after retinal surgeries.

We also found a thicker CCT in the affected eye with a greater difference in CCT in the paediatric age group. Studies have shown an increase in CCT after cataract surgery in children [24, 26, 28, 29]. The infant aphakic treatment study reported an increase in CCT after cataract extraction in both aphakic and pseudophakic with a mean CCT of aphakic eyes being higher (637 μm) than in controls (563 μm) [24]. Simon et al reported a substantial increase in CCT in aphakic and pseudophakic paediatric patients compared to paediatric or adult controls [10]. CCT values in the former ranged as high as 835 microns compared with phakic fellow eyes, which reiterates the importance of discerning if raised IOP in such situations is attributable to disease or these changes. The authors suggested revised criteria to diagnose glaucoma in such situations which may be essential in routine glaucoma practice. This study also found significant changes in age<25 years suggesting that corneal CCT changes may occur even in eyes >12 years which mandates the use of these revised criteria for diagnosing glaucoma for all eyes undergoing vitrectomy or any surgery affecting the corneal endothelium or stroma. This would also reduce false diagnosis of glaucoma and over-treatment in the paediatric age group.

Our study was a retrospective study with its inherent drawbacks. It would have been ideal to compare the increase in CCT with pre-operative CCT in the affected eye for robust results. Yet this represents a clinical scenario that is closer to the daily routine scenario that clinicians face while getting referrals for raised IOP after any surgery. Also, the variability of each eye after each surgery is expected to be extremely high suggesting that clinical decisions require a case-case customized approach. We also believe that the contralateral, surgery naïve eye that has not undergone surgery represents an indirect measure of the pre-operative CCT in the affected eye. We did not measure the viscoelastic properties of the cornea nor did we measure the CCT changes longitudinally over time. We also did not evaluate corrected IOP while diagnosing glaucoma since the relationship between CCT and glaucoma is unclear. Further, no existing published nomograms would extend into the range of CCT change as seen in this study.

In summary, this study highlights that the maximum CCT changes or corneal remodelling take place in eyes undergoing vitrectomy with maximum changes seen until age<25 years. Though aphakic and pseudophakic eyes had thicker CCT, lens status was not found to influence the changes in CCT on multivariate analysis. This highlights the need for keeping in mind the extent of CCT change in eyes undergoing surgery <25 years (or <12 years) while making a glaucoma diagnosis or deciding on treatment.

## Supporting information

**S1 Table. Comparison of various variables in affected eyes with and without pre-existing glaucoma.**
(PDF)

## Acknowledgments

Hyderabad Eye Research Foundation.

## Author Contributions

**Conceptualization:** Aparna Rao.

**Data curation:** Lisika Gawas, Aparna Rao.

**Formal analysis:** Lisika Gawas, Aparna Rao.

**Investigation:** Aparna Rao.

**Methodology:** Aparna Rao.

**Project administration:** Lisika Gawas, Aparna Rao.

**Resources:** Aparna Rao.

**Supervision:** Aparna Rao.

**Validation:** Lisika Gawas, Aparna Rao.

**Visualization:** Aparna Rao.

**Writing – original draft:** Aparna Rao.

**Writing – review & editing:** Lisika Gawas, Aparna Rao.

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
