## [Decision Letter · Decision Letter 0]

2 Feb 2021

PONE-D-20-37895

Changes in corneal thickness after vitrectomy -implications for glaucoma practice

PLOS ONE

Dear Dr. Rao,

Thank you for submitting your manuscript to PLOS ONE. After careful consideration, we feel that it has merit but does not fully meet PLOS ONE’s publication criteria as it currently stands. Therefore, we invite you to submit a revised version of the manuscript that addresses the points raised during the review process.

We look forward to receiving your revised manuscript.

Kind regards,

Michael Mimouni

Academic Editor

PLOS ONE

Journal Requirements:

Reviewers' comments:

Reviewer's Responses to Questions

**Comments to the Author**

1. Is the manuscript technically sound, and do the data support the conclusions?

Reviewer #1: Partly

Reviewer #2: Partly

2. Has the statistical analysis been performed appropriately and rigorously? 

Reviewer #1: I Don't Know

Reviewer #2: I Don't Know

3. Have the authors made all data underlying the findings in their manuscript fully available?

Reviewer #1: No

Reviewer #2: Yes

4. Is the manuscript presented in an intelligible fashion and written in standard English?

Reviewer #1: Yes

Reviewer #2: Yes

5. Review Comments to the Author

Reviewer #1: There is confusion about the methods of analysis stated in the methods and the presentation of the results. It is possible that the correct analysis was done, but that the results were presented in such a way to obscure that. I have attached details in a separate document along with a suggested alternative format for one of the tables. Either the presentation of the results should be revised or the methods should be restated to clarify what is being presented. They need to match.

Reviewer #2: The Authors provided a study on the changes of corneal thickness after vitrectomy.

The Authors reported the changes in CCT, but to help understand the pathogenesis of these changes, it will be useful to correlate the change in CCT to the type and indications for the surgery. Since they stated that they have collected these data, I think performing this analysis could be valuable to complete their study.

The manuscript is generally well written; however, I recommend an English revision for its consideration as publishable.

To follow some specific comments:

- Title: ‘Implications for glaucoma practice’. The authors did not explain how the glaucoma practice will change after their results;

- Methods section needs to be revised, authors need to be more precise and describe in detail the procedure they used for measuring IOP and the surgical techniques.

- Your criteria for the diagnosis of glaucoma are vague, are we talking about ocular hypertension or glaucoma? You need to be consistent and to follow specific diagnostic criteria.

- ‘standard glaucoma practice guidelines’. Which ones? Please reference.

- Stats: was the normality of the data checked before the use of t-test? If the case, please report it.

- Tables. Data reported are not easy to read, especially Table 2. Please be clearer.

6. PLOS authors have the option to publish the peer review history of their article (what does this mean?). If published, this will include your full peer review and any attached files.

Reviewer #1: **Yes: **Sandra Stinnett

Reviewer #2: No

---

## [Author Response · Author response to Decision Letter 0]

18 Feb 2021

5. Review Comments to the Author

Reviewer #1: There is confusion about the methods of analysis stated in the methods and the presentation of the results. It is possible that the correct analysis was done, but that the results were presented in such a way to obscure that. I have attached details in a separate document along with a suggested alternative format for one of the tables. Either the presentation of the results should be revised or the methods should be restated to clarify what is being presented. They need to match.

Answer: We have provided the tables in the same way as stated in methods, Page 6-For ex- comparison of normal versus affected eye in methods with table1,2, comparison of Anterior and posterior vitrectomy eyes comparisons in table 2 and so forth. Also, we have provided both absolute CCT and ∆CCT as detailed in the methods section again. We have still changed table format as suggested while keeping details in table 2 for benefit of readers. Table 3 and Table S1 are also those that are stated in the methods section, page 6. We have made sure we have given all relevant data in the study for the benefit of readers rather than obscure any detail.

Reviewer #2: The Authors provided a study on the changes of corneal thickness after vitrectomy.

The Authors reported the changes in CCT, but to help understand the pathogenesis of these changes, it will be useful to correlate the change in CCT to the type and indications for the surgery. Since they stated that they have collected these data, I think performing this analysis could be valuable to complete their study.

Answer: We thank the reviewer for this suggestion and we agree with the reviewer that indication of surgery may be also looked at. We would like to clarify that we looked at type of vitrectomy done- anterior versus posterior rather than look at indications of surgery which may be varied and would dilute the objectives of this study (say for example surgery for dislocated Lens versus DR versus ELM peeling +PPV- there would be too many surgical variables all of which cannot be assessed statistically for each group and would never conclude with a meaningful message with several subgroup analysis). Yet, we did stratify the CCT changes by indication of surgery like PDR and TRD and did not note any significant change between the two indications and therefore have not added that in tables or text.

The manuscript is generally well written; however, I recommend an English revision for its consideration as publishable.

Answer: The manuscript has been edited by Grammarly for correcting typos and grammar, as suggested.

To follow some specific comments:

- Title: ‘Implications for glaucoma practice’. The authors did not explain how the glaucoma practice will change after their results;

Answer: The implications of a change in CCT reflecting as “raised IOP” after VR surgeries or after paediatric surgeries have been discussed in detail in two separate paragraphs in the discussion, page 9 and 10. Basically, we feel that a false alarm of glaucoma is avoided if we know that the changed CCT can falsely present as raised IOP. We thank the reviewer and have added this in discussion highlighting this implication in glaucoma practice, as suggested, Page 11 and 12

.

- Methods section needs to be revised, authors need to be more precise and describe in detail the procedure they used for measuring IOP and the surgical techniques.

Answer: Being a retrospective study, we could not have blinded the trained optometrists in the study. The methods of IOP measurement and CCT are given in the methods section which have been modified as per suggestions. 

 Your criteria for the diagnosis of glaucoma are vague, are we talking about ocular hypertension or glaucoma? You need to be consistent and to follow specific diagnostic criteria.

Answer: We thank the reviewer for this observation. We have stated in the methods that for this retrospective study, we diagnosed secondary glaucoma in the event of a raised IOP >21mm Hg which is as per standard glaucoma diagnostic guidelines. We have also stated that disc/field changes at presentation were not mandatory for the diagnosis. So the reviewer is correct that initially they may seem as ocular hypertension; yet it is logical to understand disc/field changes would not appear immediately after raised IOP anyway. This is the reason a longitudinal prospective study may identify how many turned out be glaucoma versus ow many remained OHT wich wa not possible for this retrospective study. The entire premise of the study was to see if how this diagnostic criterion used in clinics for eyes with secondary glaucoma fares, keeping in mind the CCT changes after surgery. This mimics routine cincial scenarios and does not compare to matched /controlled prospective trials with strict definitions. The same has been discussed extensively in the results and the discussion. For a prospective study, we can revise the diagnosis based on disc/field changed over time. 

- ‘standard glaucoma practice guidelines’. Which ones? Please reference.

Answer: The sentences has been rephrased, as suggested. As institutional protocol, we follow AAO and ISGEO guidelines among many others for glaucoma management.

- Stats: was the normality of the data checked before the use of t-test? If the case, please report it.

- Tables. Data reported are not easy to read, especially Table 2. Please be clearer.

Answer: Normality was indeed checked as per norms for systematic statistical analysis and parametric tests used only for normative data. Table 2 highlights the mean CCT comparisons in affected and normal eye in different age groups and between eyes that underwent anterior and posterior route of vitrectomy. This format was important to highlight that changes specific to a particular age group and particular route of surgery. This table also highlights one of the most important message of the study that is- greater ∆CCT in eyes presenting at an age <25 years, which could not have been possible had we presented differently. We have changed the tables with incorporation of all changes suggested by the reviewer. We further clarify that ∆CCT was calculated by seeing them in each eye rather than averaging CCT in affected and normal eye and seeing difference. This was because we realise that the difference between two eyes would be very very variable for each patient and also possibly by each surgery (wit a lot of surgical variables). This vast variability will not be highlighted had we averaged all affected eye and all normal eyes and then seen the difference, which would have hidden and smoothened out the different variability with a single mean. This also would have been a wrong way of measuring the difference caused by each surgery or in different age groups. The number of eyes are given in table 3 clearly for each group. We have incorporated all suggestions for depicting the results in the format given by the reviewer and thank the reviewer for such a nice suggestion for explaining our results in a better way.

We also thank the reviewer for the insightful suggestion on the table format, which have been changed. We have also added number in the tables as suggested and thank the reviewer for pointing this out. We would like to clarify that there were 127 eyes of 120 patients with in eye that has absolute at presentation being excluded. Yes, the reviewer is correct that paired t test ideally requires bilateral cases only which indeed is correct and has been added in the table. Also the CCT doff therefore could be done in 120 eyes only with both eyes included only. We thank the reviewer for useful tips to highlight in the tables for a better understanding and we appreciate and have incorporated the changes. ________________________________________________________________________________

---

## [Decision Letter · Decision Letter 1]

10 Mar 2021

PONE-D-20-37895R1

Changes in corneal thickness after vitrectomy -implications for glaucoma practice

PLOS ONE

Dear Dr. Rao,

Thank you for submitting your manuscript to PLOS ONE. After careful consideration, we feel that it has merit but does not fully meet PLOS ONE’s publication criteria as it currently stands. Therefore, we invite you to submit a revised version of the manuscript that addresses the points raised during the review process.

We look forward to receiving your revised manuscript.

Kind regards,

Michael Mimouni

Academic Editor

PLOS ONE

Journal Requirements:

Reviewers' comments:

Reviewer's Responses to Questions

**Comments to the Author**

1. If the authors have adequately addressed your comments raised in a previous round of review and you feel that this manuscript is now acceptable for publication, you may indicate that here to bypass the “Comments to the Author” section, enter your conflict of interest statement in the “Confidential to Editor” section, and submit your "Accept" recommendation.

Reviewer #1: (No Response)

Reviewer #2: All comments have been addressed

2. Is the manuscript technically sound, and do the data support the conclusions?

Reviewer #1: Yes

Reviewer #2: Yes

3. Has the statistical analysis been performed appropriately and rigorously? 

Reviewer #1: Yes

Reviewer #2: N/A

4. Have the authors made all data underlying the findings in their manuscript fully available?

Reviewer #1: Yes

Reviewer #2: Yes

5. Is the manuscript presented in an intelligible fashion and written in standard English?

Reviewer #1: Yes

Reviewer #2: Yes

6. Review Comments to the Author

Reviewer #1: In the results section under “affected eye comparisons, the authors state: “The average CCT in the affected eye was significantly higher than the unaffected contralateral control eye (p<0.0001), Table 1, Figure 1.”

These are my comments from my first review:

3. In table 1, they present the IOP, visual acuity and CCT in two columns without presenting the average difference. It is the average difference that is being tested, not the difference in means between eyes. It is fine to show the means for each eye, but they need to also show the average difference. The layout in the table makes it appear that they have treated this as independent data. This is further unclear because they do not put N’s in the table. They need to show the number of eyes.

4. Their expression of the results for Table 1 also makes it appear that they did not analyze the average difference: “The average CCT in the affected eye was significantly higher than the unaffected contralateral control eye (p<0.0001).”

As I expressed in the original review, this is not a correct statement. You are not comparing the difference in means between the affected eye and the unaffected eye, as this sentence implies. This makes the reader think that you have carried out an independent t-test. I see that you have added a footnote to table 1 that indicates that a paired t-test was carried out. However, we still don’t know what the average difference was since that is not shown in the tables. Since only 120 eyes are being compared yet we see 127 for the affected eye, even the means in the tables are not reflective of what is being compared. The correct way to express that a paired t-test was done (and not an independent test) is to say that the average difference between eyes was significant and tell the reader what that was. You could leave the table the way it is if you express in the text the average difference, standard deviation of the difference along with the p-value you already have given.

The authors have made changes to Table 2 which make it easier to understand. It would be helpful to make the same changes to Table 1. Then the difference between eyes would be perfectly clear.

Reviewer #2: The Authors put a lot of efforts to improve their manuscript, and they have successfully replied to all my comments. Nevertheless, a lot of typos and formatting mistakes are still present in the manuscript (just too many to list them here) and need to be fixed. Please check this aspect carefully.

7. PLOS authors have the option to publish the peer review history of their article (what does this mean?). If published, this will include your full peer review and any attached files.

Reviewer #1: No

Reviewer #2: No

---

## [Author Response · Author response to Decision Letter 1]

11 Mar 2021

To,

The Editor,

Dear Sir/Madam,

We hereby submit our revised manuscript “Changes in corneal thickness after vitrectomy -implications for glaucoma practice” for review for publication in your journal. We have professionally edited the manuscript for grammar and typo errors, as suggested. We would welcome any more suggestions or queries which can further improve our manuscript. We have included all suggestions made by reviewers for wider readership and easy understanding which has improved the impact of the manuscript considerably. 

All the authors have contributed equally towards preparation of the manuscript and have no financial or proprietary interest in the products used in the study. We also declare that this article has not been published previously or under review with any other journal.

a) We have made available all supplemental files/data for the reader’s benefit and can make all clinical raw data available for public repository on request. 

b) All data are available with the corresponding author or the institute database/IRB which can be shared after third party inclusion for specific interested readers for data sharing by the patients. 

Thanking you

Reviewer #1: In the results section under “affected eye comparisons, the authors state: “The average CCT in the affected eye was significantly higher than the unaffected contralateral control eye (p<0.0001), Table 1, Figure 1.”

These are my comments from my first review:

3. In table 1, they present the IOP, visual acuity and CCT in two columns without presenting the average difference. It is the average difference that is being tested, not the difference in means between eyes. It is fine to show the means for each eye, but they need to also show the average difference. The layout in the table makes it appear that they have treated this as independent data. This is further unclear because they do not put N’s in the table. They need to show the number of eyes.

4. Their expression of the results for Table 1 also makes it appear that they did not analyze the average difference: “The average CCT in the affected eye was significantly higher than the unaffected contralateral control eye (p<0.0001).”

As I expressed in the original review, this is not a correct statement. You are not comparing the difference in means between the affected eye and the unaffected eye, as this sentence implies. This makes the reader think that you have carried out an independent t-test. I see that you have added a footnote to table 1 that indicates that a paired t-test was carried out. However, we still don’t know what the average difference was since that is not shown in the tables. Since only 120 eyes are being compared yet we see 127 for the affected eye, even the means in the tables are not reflective of what is being compared. The correct way to express that a paired t-test was done (and not an independent test) is to say that the average difference between eyes was significant and tell the reader what that was. You could leave the table the way it is if you express in the text the average difference, standard deviation of the difference along with the p-value you already have given.

The authors have made changes to Table 2 which make it easier to understand. It would be helpful to make the same changes to Table 1. Then the difference between eyes would be perfectly clear.

Answer: We had clarified that ∆CCT was calculated by seeing them in each eye rather than averaging CCT in affected and normal eye. Yet, the mean CCT between normal and affected eyes were compared by paired t test as written in methods and reflected in Table1. We had also elaborated the reason for choosing this method and would like to add that ∆CCT is an indirect measure of the change in same patient and not a direct longitudinal measurement, meaning the control eye CCT reflects the possible baseline CCT of the affected eye, while CCT of the affected eye reflects the change induced by surgery (This is already detailed in the text in methods and discussion). So while we understand the suggestion of average differences being included in the text, this would mean a different clinical message which is very confusing readers and take away key messages and highlights. We also point here that ,as per earlier suggestions, we included N in modified table 1 earlier itself and also changed the format of the table1 along with table 2. 

Let us in detail explain the above by examples

Lets say we have the below CCT from 3 pts

Affected eye Normal eye

1. 602 567

2. 602 489

3. 680 and 602 (Ou undergoing VR sx)

Here affected eye n=4, control eye n=2-----t test run between mean of Affected and normal eyes that is 617 and 528, respectively, (The N also is clear as to why we have 127 affected and 120 controls-method in revised text has included how many were both eyes) 

But ∆CCT is calculated for 2 eyes only(pt 1 and 2)- ∆CCT would be calculated from diff in pt and 2 (35 and 113). This happens because the CCT in normal also is highly variable-so calculating all affected eyes and normal as one may not reflect the indirect CCT changes, as described above. 

We clarify that ∆CCT was calculated by seeing them in each eye rather than averaging CCT in affected and normal eye and seeing difference in their means. Yet the Paired t test was run between means which is reflected in table1. This was because we realise that the difference between two eyes would be far too variable for each patient and also possibly by each surgery (with a lot of surgical variables). This vast variability will not be highlighted had we averaged all affected eye and all normal eyes and then seen the difference, which would have hidden and smoothened out the different variability with a single mean. This also would have been a wrong way of measuring the difference caused by each surgery or in different age groups.

Reviewer #2: The Authors put a lot of efforts to improve their manuscript, and they have successfully replied to all my comments. Nevertheless, a lot of typos and formatting mistakes are still present in the manuscript (just too many to list them here) and need to be fixed. Please check this aspect carefully.

Answer: The manuscript has now been edited by Grammarly but there may be overlapping of words that happen with diff Microsoft versions we noticed in the PDF. 

Earlier reviewer comments

5. Review Comments to the Author

Reviewer #1: There is confusion about the methods of analysis stated in the methods and the presentation of the results. It is possible that the correct analysis was done, but that the results were presented in such a way to obscure that. I have attached details in a separate document along with a suggested alternative format for one of the tables. Either the presentation of the results should be revised or the methods should be restated to clarify what is being presented. They need to match.

Answer: We have provided the tables in the same way as stated in methods, Page 6-For ex- comparison of normal versus affected eye in methods with table1,2, comparison of Anterior and posterior vitrectomy eyes comparisons in table 2 and so forth. Also, we have provided both absolute CCT and ∆CCT as detailed in the methods section again. We have still changed table format as suggested while keeping details in table 2 for benefit of readers. Table 3 and Table S1 are also those that are stated in the methods section, page 6. We have made sure we have given all relevant data in the study for the benefit of readers rather than obscure any detail.

Reviewer #2: The Authors provided a study on the changes of corneal thickness after vitrectomy.

The Authors reported the changes in CCT, but to help understand the pathogenesis of these changes, it will be useful to correlate the change in CCT to the type and indications for the surgery. Since they stated that they have collected these data, I think performing this analysis could be valuable to complete their study.

Answer: We thank the reviewer for this suggestion and we agree with the reviewer that indication of surgery may be also looked at. We would like to clarify that we looked at type of vitrectomy done- anterior versus posterior rather than look at indications of surgery which may be varied and would dilute the objectives of this study (say for example surgery for dislocated Lens versus DR versus ELM peeling +PPV- there would be too many surgical variables all of which cannot be assessed statistically for each group and would never conclude with a meaningful message with several subgroup analysis). Yet, we did stratify the CCT changes by indication of surgery like PDR and TRD and did not note any significant change between the two indications and therefore have not added that in tables or text.

The manuscript is generally well written; however, I recommend an English revision for its consideration as publishable.

Answer: The manuscript has been edited by Grammarly for correcting typos and grammar, as suggested.

To follow some specific comments:

- Title: ‘Implications for glaucoma practice’. The authors did not explain how the glaucoma practice will change after their results;

Answer: The implications of a change in CCT reflecting as “raised IOP” after VR surgeries or after paediatric surgeries have been discussed in detail in two separate paragraphs in the discussion, page 9 and 10. Basically, we feel that a false alarm of glaucoma is avoided if we know that the changed CCT can falsely present as raised IOP. We thank the reviewer for ths suggestion and have added this in discussion highlighting this implication in glaucoma practice, as suggested, Page 11 and 12

.

- Methods section needs to be revised, authors need to be more precise and describe in detail the procedure they used for measuring IOP and the surgical techniques.

Answer: Being a retrospective study, we could not have blinded the trained optometrists in the study. The methods of IOP measurement and CCT are given in the methods section which have been modified as per suggestions. 

 Your criteria for the diagnosis of glaucoma are vague, are we talking about ocular hypertension or glaucoma? You need to be consistent and to follow specific diagnostic criteria.

Answer: We thank the reviewer for this observation. We have stated in the methods that for this retrospective study, we diagnosed secondary glaucoma in the event of a raised IOP >21mm Hg which is as per standard glaucoma diagnostic guidelines. We have also stated that disc/field changes at presentation were not mandatory for the diagnosis. So the reviewer is correct that initially they may seem as ocular hypertension; yet it is logical to understand disc/field changes would not appear immediately after raised IOP anyway. This is the reason a longitudinal prospective study may identify how many turned out be glaucoma versus ow many remained OHT wich wa not possible for this retrospective study. The entire premise of the study was to see if how this diagnostic criterion used in clinics for eyes with secondary glaucoma fares, keeping in mind the CCT changes after surgery. This mimics routine clinical scenarios and does not compare to matched /controlled prospective trials with strict definitions. The same has been discussed extensively in the results and the discussion. For a prospective study, we can revise the diagnosis based on disc/field changed over time. 

- ‘standard glaucoma practice guidelines’. Which ones? Please reference.

Answer: The sentences has been rephrased, as suggested. As institutional protocol, we follow AAO and ISGEO guidelines among many others for glaucoma management.

- Stats: was the normality of the data checked before the use of t-test? If the case, please report it.

- Tables. Data reported are not easy to read, especially Table 2. Please be clearer. 

Answer: Normality was indeed checked as per norms for systematic statistical analysis and parametric tests used only for normative data. Table 2 highlights the mean CCT comparisons in affected and normal eye in different age groups and between eyes that underwent anterior and posterior route of vitrectomy. This format was important to highlight that changes specific to a particular age group and particular route of surgery. This table also highlights one of the most important message of the study that is- greater ∆CCT in eyes presenting at an age <25 years, which could not have been possible had we presented differently. We have changed the tables with incorporation of all changes suggested by the reviewer. We further clarify that ∆CCT was calculated by seeing them in each eye rather than averaging CCT in affected and normal eye and seeing difference. This was because we realise that the difference between two eyes would be far too variable for each patient and also possibly by each surgery (with a lot of surgical variables). This vast variability will not be highlighted had we averaged all affected eye and all normal eyes and then seen the difference, which would have hidden and smoothened out the different variability with a single mean. This also would have been a wrong way of measuring the difference caused by each surgery or in different age groups. The number of eyes are given in table 3 clearly for each group. We have incorporated all suggestions for depicting the results in the format given by the reviewer and thank the reviewer for such a nice suggestion for explaining our results in a better way.

We also thank the reviewer for the insightful suggestion on the table format, which have been changed. We have also added number in the tables as suggested and thank the reviewer for pointing this out. We would like to clarify that there were 127 eyes of 120 patients with in eye that has absolute at presentation being excluded. Yes, the reviewer is correct that paired t test ideally requires bilateral cases only which indeed is correct and has been added in the table. Also the CCT doff therefore could be done in 120 eyes only with both eyes included only. We thank the reviewer for useful tips to highlight in the tables for a better understanding and we appreciate and have incorporated the changes. ________________________________________

---

## [Decision Letter · Decision Letter 2]

29 Mar 2021

Changes in corneal thickness after vitrectomy -implications for glaucoma practice

PONE-D-20-37895R2

Dear Dr. Rao,

We’re pleased to inform you that your manuscript has been judged scientifically suitable for publication and will be formally accepted for publication once it meets all outstanding technical requirements.

Kind regards,

Michael Mimouni

Academic Editor

PLOS ONE

Additional Editor Comments (optional):

Reviewers' comments:

Reviewer's Responses to Questions

**Comments to the Author**

1. If the authors have adequately addressed your comments raised in a previous round of review and you feel that this manuscript is now acceptable for publication, you may indicate that here to bypass the “Comments to the Author” section, enter your conflict of interest statement in the “Confidential to Editor” section, and submit your "Accept" recommendation.

Reviewer #1: (No Response)

Reviewer #2: All comments have been addressed

2. Is the manuscript technically sound, and do the data support the conclusions?

Reviewer #1: Yes

Reviewer #2: Yes

3. Has the statistical analysis been performed appropriately and rigorously? 

Reviewer #1: (No Response)

Reviewer #2: Yes

4. Have the authors made all data underlying the findings in their manuscript fully available?

Reviewer #1: Yes

Reviewer #2: Yes

5. Is the manuscript presented in an intelligible fashion and written in standard English?

Reviewer #1: Yes

Reviewer #2: Yes

6. Review Comments to the Author

Reviewer #1: I need to just let this go. I am not getting through to the authors. This is their explanation below which is even more confusing because they say that the "paired t-test was run between means." You can't ran a paired t-test between means. That's my point. A paired t-test is run on the average difference between the individual eyes. Nowhere do they say what the average difference is, only what the mean for each eye is. Suppose I wanted to use their results to power another study, I would want to know the average difference and the standard deviation of the difference. The fact that they say that the difference between individual eyes would be too variable is another point FOR (not against) using a paired t-test (the difference between eyes means that each person is their own control).

We clarify that ∆CCT was calculated by seeing them in each eye rather than averaging

CCT in affected and normal eye and seeing difference in their means. Yet the Paired t

test was run between means which is reflected in table1. This was because we realise

that the difference between two eyes would be far too variable for each patient and

also possibly by each surgery (with a lot of surgical variables). This vast variability will

not be highlighted had we averaged all affected eye and all normal eyes and then seen

the difference, which would have hidden and smoothened out the different variability

with a single mean. This also would have been a wrong way of measuring the

difference caused by each surgery or in different age groups

Reviewer #2: The Authors have addressed all the comments and revised the manuscript accordingly.

I have no additional comments.

7. PLOS authors have the option to publish the peer review history of their article (what does this mean?). If published, this will include your full peer review and any attached files.

Reviewer #1: No

Reviewer #2: No

---

## [Editor Report · Acceptance letter]

5 Apr 2021

PONE-D-20-37895R2 

Changes in corneal thickness after vitrectomy -implications for glaucoma practice 

Dear Dr. Rao:

I'm pleased to inform you that your manuscript has been deemed suitable for publication in PLOS ONE. Congratulations! Your manuscript is now with our production department. 

Kind regards, 

on behalf of

Dr. Michael Mimouni 

Academic Editor

PLOS ONE